# A North-West London Experience of the Impact of Treatment Related Toxicity on Clinical Outcomes of Elderly Patients with Germ Cell Tumors

**DOI:** 10.3390/cancers14204977

**Published:** 2022-10-11

**Authors:** Anand Sharma, Laura Morrison, Marina Milic, Aruni Ghose, Andrew Gogbashian, Nikhil Vasdev, Samita Agarwal, Ben Pullar, Gordon Rustin

**Affiliations:** 1Department of Medical Oncology, Mount Vernon Cancer Centre, London HA6 2RN, UK; 2Department of Radiology, Mount Vernon Cancer Centre, Paul Strickland Scanner Centre, London HA6 2RN, UK; 3Hertfordshire and Bedfordshire Urological Cancer Centre, Department of Urology, Lister Hospital, East and North Herts NHS Trust, Stevenage SG1 4AB, UK; 4School of Life and Medical Sciences, University of Hertfordshire, Hatfield AL10 9EU, UK; 5Department of Histopathology, Lister Hospital, East and North Herts NHS Trust, Stevenage SG1 4AB, UK

**Keywords:** testicular cancer, elderly, chemotherapy, complications, clinical outcomes

## Abstract

**Simple Summary:**

Germ cell testicular cancer is seen in men aged 15–35. Incidence of this cancer in the elderly aged > 45 is less than 10%. The aim of our retrospective study is to understand the treatment-related toxicity in the elderly population, the chemotherapy regimens used, and its tolerability in this age group. Incidence of non-seminomatous germ cell cancer was 42% and incidence of seminoma was 58% in this cohort of patients. We used chemotherapy regimens, similar to those used in the younger population; platinum-based regimens were the backbone of the treatment algorithm. Outcomes were similar to the population group aged 15–35, and around 96% of the population were alive 5 years after and beyond. Toxicity was more common in this age group, and careful consideration should be made for dose modification and appropriate use of supportive therapies.

**Abstract:**

Background/Aim: The occurrence of germ cell tumour (GCT) in the elderly is rare, with scarce data available. The aim of this study was to understand the clinical outcomes of patients with GCT in patients aged > 45 years. Materials and Methods: A retrospective study was conducted in a large tertiary cancer centre in north-west London. Between 1 January 2003 and 31 March 2022, 108 cases of GCT in men aged > 45 years were identified and treated at the Mount Vernon Cancer Centre. The median age at diagnosis was 54 years (range = 45–70 years). Results: The 5-year survival rate of all patients was 96%, and the toxicity profile was similar to the younger age group. Conclusion: Older patients with GCT are able to tolerate chemotherapy; however, care must be taken to prevent life-threatening complications using appropriate dose modification.

## 1. Introduction

Germ cell tumours (GCTs) are the most common malignancy in patients aged between 15 and 35. However, cases in older patients remain relatively rare with <10% of the total number of germ cell tumours being diagnosed in this age group [1]. GCTs can be classified into two major categories according to their histology: classical seminoma and non-seminomatous or mixed GCTs (NSGCT). The second category can include embryonal carcinoma, yolk sac tumour, choriocarcinoma, teratoma or a mixture of histological subtypes [2,3]. There is some evidence to suggest that the ratio of seminomatous to non-seminomatous tumours alter with increasing age, and that seminomatous tumours become the dominant subtype in men aged over 35 years [4].

Metastatic GCTs can further be subdivided in terms of their risk of relapse into favourable, intermediate and poor risk; according to the combination of tumour size, histology, distribution of disease and the level of the tumour markers; alfa feto-protein (AFP; human chorionic gonadotrophin hormone (HCG); and lactate dehydrogenase (LDH). This risk stratification assists in prognostication, decision making in the clinic, and helps to determine chemotherapy regime options and the schedule for ongoing follow up [5].

Metastatic GCTs have high cure rates of up to 90% with first line chemotherapy. Platinum-based chemotherapy is the backbone of most regimens in current practice. Various regimens are used to treat GCTs such as BEP (bleomycin, etoposide, and cisplatin), POMB/ACE (cisplatin, vincristine, methotrexate, bleomycin/actinomycin-D, cyclophosphamide, and etoposide), GEM/TIP (gemcitabine, paclitaxel, ifosfamide, and cisplatin), GAMEC (actinomycin-D, methotrexate, cisplatin and etoposide), TE/TP (paclitaxel, etoposide/paclitaxel, and cisplatin) [6]. At our centre, seminomas are treated with carboplatin (AUC10), which is a well-tolerated regimen [7,8]. Because of the relatively low number of patients over the age of 45 with GCT, there is little information about the tolerability and toxicity profiles in this cohort of patients [9]. While generally well tolerated in the younger population [10], there is anecdotal evidence that toxicity may be increased in an older patient cohort, particularly in relation to cisplatin [11,12]. The combination of drugs in GCTs can result in significant toxicity which may lead to permanent changes in hearing [13], renal function, lung function, and neurology as a result of peripheral neuropathy [14].

Increasing toxicities may result in the completion of fewer cycles of chemotherapy, or the need for dose reductions, which could in turn affect cure rates and overall survival. A retrospective analysis performed with a similar cohort at MSKCC identified 50 patients aged over 50 years receiving chemotherapy for GCTs. They found high rates of discontinuation (60% of patients) across regimens and in those receiving BEP chemotherapy, 72% required a change to an alternative regime. EP and VIP seemed to be better tolerated, with greater than 95% completing treatment as planned. This high complication rate differs from published data and phase three studies which are predominantly carried out in younger men. While some of this reduced tolerance may be related to co-morbidities and already reduced functional ability, age can have an adverse effect on organ function, impairing the metabolism [15] and clearance of chemotherapy agents [16].

## 2. Patients and Methods

An internal database was used to identify all patients diagnosed with testicular germ cell tumours between 1 January 2003 and 31 March 2022, aged over 45 years at the time of diagnosis and referred to Mount Vernon Cancer Centre (MVCC), London, for systemic chemotherapy. Appropriate approval was sought from the hospital ethics committee. Patients over 45 comprised 20% of the total number of patients being referred within the given time period, giving a total number of 108 patients aged over 45 at the time of diagnosis and initiation of chemotherapy (104 metastatic and 4 adjuvant).

Data were collected retrospectively from the notes and electronic records and recorded in a database.

For each patient we assessed the stage at diagnosis using the seventh edition of the TNM classification. This uses a combination of computed tomographic (CT) imaging of the chest, abdomen and pelvis, as well as serum tumour markers AFP, β-HCG, and LDH. We used both CT and tumour markers where available.

The histology was recorded from the pathology reports of orchidectomy or biopsy samples and correlated with the World Health Organisation (WHO) classification of GCTs. We also recorded the international germ cell consensus risk classification (IGCC) for all patients where available. The number of full chemotherapy cycles completed was identified using the patient’s notes and chemocare records and was recorded alongside the need for a dose reduction or the discontinuation of treatment. If discontinuation of treatment was required, the reasons for this were also documented. Dose reductions were considered for haematological toxicity and on the basis of clinical assessment prior to chemotherapy administration. Patients were able to receive blood products as required during treatment, as per local protocols.

All patients underwent blood laboratory tests including full blood count, urea and electrolytes, liver function and serum tumour markers prior to each cycle of treatment. They were assessed clinically and routinely asked about breathlessness and cough as possible symptoms of bleomycin-induced pulmonary toxicity.

Toxicities were assessed using a locally designed questionnaire based on the Common Terminology Criteria for Adverse Events (CTCAE) Version 4.1 and recorded during clinical assessments while the patients were receiving chemotherapy and in follow up clinic appointments.

Response to treatment was assessed using serum tumour markers and CT imaging according to Response Evaluation Criteria in Solid Tumours (RECIST) 1.1. Relapse was also documented according to biochemical and CT evidence of disease recurrence.

Following completion of their chemotherapy, patients were routinely followed for 10 years. This comprised three monthly reviews in the first year, four monthly in the second year, six monthly from year 3 to 5, and annually thereafter. Serum tumour markers were measured at each clinic appointment. CT imaging was performed at the end of treatment and only repeated if in a clinical trial, the end of treatment scan was abnormal, or if clinically indicated.

CT imaging and blood results were used to evaluate progression free survival, relapse rates, 5-year survival, and overall survival in this cohort.

Efficacy was assessed in all patients who received at least 50% of the planned chemotherapy. OS and PFS were estimated using the Kaplan–Meier method. Safety was assessed in all patients; summary statistics were provided for baseline demographics, disease characteristics, and AEs. Results are based on interim analysis of data, with a cut-off date of 31 March 2022; patients continue to be observed for long-term outcomes. The primary end point was OS, defined as the number of patients alive at the cut-off date of 31 March 2022.

## 3. Results

One hundred and eight patients aged over 45 years at the time of diagnosis and initiation of systemic chemotherapy were identified and included in this study following their systemic chemotherapy. The mean age was 54 (range 45–70). Thirty one percent (*n* = 33) of all patients were aged between 50 and 54 years, 27% were aged between 45–49 years (*n* = 30) and 20% aged between 55–59 years (*n* = 21). The remaining 22% were aged over 60 years (*n* = 24) (Table 1).

Classical seminoma was seen in 58% of the cohort, with 42% having a non-seminomatous or mixed germ cell picture. Of those with NSGCT, thirty-two had a mixed GCT; eight had a teratoma, two had a choriocarcinoma, two an embryonal, and one a yolk sac tumour.

The most frequently prescribed first-line regimen was BEP, with 43% (*n* = 47) receiving this regime, followed by Carboplatin AUC 10 (17%), POMB/ACE (10%), Etoposide and cisplatin (EP) (10%), and Carboplatin AUC 7 (4%) (Table 2). In the second line, four patients received Gem/TIP and Paclitaxel + Etoposide/Paclitaxel + Cisplatin (TE/TP) (Table 2).

The majority of patients (77%) were stage II or greater at the time of diagnosis. Thirteen patients (22%) had stage I disease, thirty-one (53%) stage II, seven (12%) stage III, and seven (12%) stage IV disease at diagnosis. Of the 13 patients who had stage I disease, 11 relapsed on surveillance and 2 were seminomas (who had Carboplatin AUC7 previously).

Forty-one patients completed their chemotherapy as originally planned at the outset of treatment, with three further patients requiring a delay in treatment due to haematological toxicity, ten patients required a modification to their regime such as dose reduction or omission of a dose of chemotherapy, and five of the patients receiving BEP had the bleomycin omitted for part of the treatment. Three of the patients on BEP developed lung toxicity apparent on clinical assessment and CT imaging. Fifteen patients had a documented neutropenia during treatment, with six of these developing fevers and requiring neutropenic sepsis treatment according to local procedures at the admitting hospital. Thirteen patients developed sensory neuropathy mainly affecting the feet and lower legs. Five patients developed tinnitus, although none of these cases were worse than grade 1 (Table 3).

Four patients did not complete the chemotherapy as planned: one due to death from a pulmonary embolism, and the remaining three due to toxicities.

Five patients died during our study follow up. Only one patient died while on chemotherapy from a pulmonary embolism after he had a sustained a gastrointestinal bleed secondary to the anti-coagulation for the embolism. A second patient died of an unrelated cause on a background of significant underlying ischaemic heart disease, NYHA class 3, previous SCC of the anus, and COPD (22 months after chemotherapy). A third patient died from ischaemic heart disease after completion of follow up (10 years post diagnosis). The fourth patient died of Alzheimer’s related complications, which was diagnosed 7 years after completion of chemotherapy. There are no available records for the cause of death for the fifth patient.

Sixty-nine percent (41/59) of patients experienced one or more complications from chemotherapy, of which 15 were CTCAE grade 3 or above (Table 3). The toxicities are shown in Table 3. The commonest treatment related toxicities were neutropenia (19%), neuropathy (19%), neutropenic fever (10%), thromboembolism (10%) and tinnitus (10%).

Five-year survival was 96%, with one hundred and three patients surviving for five years. Overall survival across the cohort at the time of data cut-off was 92% (Figure 1). Only one patient died during chemotherapy as a direct result of complications from his disease.

## 4. Discussion

There is limited data for patients aged over 45 years regarding chemotherapy toxicity. Anecdotal evidence suggests patients over 50 years may struggle with toxicities related to chemotherapy regimens used to treat germ cell cancer [17]. While this study included a relatively small number of patients, it is clear that those aged 45 years or more can tolerate significant doses of chemotherapy in combination regimens identical to those used in a younger cohort. The small number of patients identified over a fifteen-year period reflects the greater prevalence of germ cell tumours in a younger population, and its rarity in those over the age of 45.

Survival rates in this population remain good, with the majority (96%) surviving at least 5 years.

Toxicity rates were similar to those observed in the younger population [10], although pulmonary toxicity [18,19] remains a concern in the older population [20,21], especially among those with a history of smoking [22]. In our cohort, 14% of patients receiving BEP required an omission of bleomycin due to pulmonary toxicity. The commonest side effect across all regimens documented was neutropenia, with 10% of patients going on to develop a febrile neutropenia [23]. Sensory neuropathy, a well-known side effect of platinum chemotherapy, was also noted in a significant number of the patients; this may exacerbate underlying mobility issues in older patients [24].

While age may not in itself be a prognostic factor in terms of survival [20], it is clear that underlying comorbidities alongside age can leave older patients more susceptible to toxicities associated with chemotherapy [14,25], and treating oncologists will need to be aware of the potential need for dose reductions and treatment delays. Given that previous studies have shown a higher incidence of cancer-specific mortality in older patients with GCTs [26,27], it is important to be able to offer them dose-intense chemotherapy regimens to attempt to improve their overall survival. It is beyond the scope of this audit to assess the impact of dose reductions or delays on overall survival and relapse rates.

We acknowledge the limitations of this study, which are mainly integral to its retrospective design and age selection. Previous reports have included patients aged 50 and above in a similar analysis. However, due to the nature of this disease, and tolerability of high dose chemotherapy in men aged 40 and above, we consider 45 to be a good marker for assessing the effect of age on tolerability in elderly germ cell cancer patients. Another limitation is the use of different chemotherapy regimens and lack of standardization in comparing the data in different subgroups.

## 5. Conclusions

Intensive chemotherapy regimens are tolerable in patients over the age of 45 years, however care must be taken to identify potential toxicities and adjust doses and regimens accordingly. Older patients should be offered dose-intense chemotherapy with curative intent. The spectrum of outcomes and adverse events is similar in the elderly patients.

## Figures and Tables

**Figure 1 cancers-14-04977-f001:**
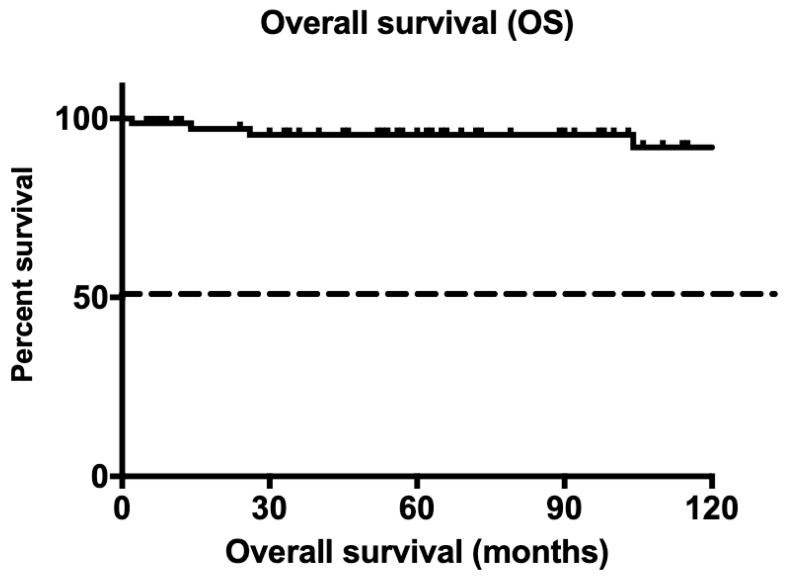
Kaplan Meir curve for overall survival.

**Table 1 cancers-14-04977-t001:** Year of diagnosis and demographics.

Year of Diagnosis	Number of Patients (108)	Average Age	Diagnosis	
			Seminoma (63)	Non-Seminoma (45)
2003–2006	16	52	9	7
2007–2010	20	53	15	5
2011–2014	21	53	10	11
2015–2018	26	54	16	10
2019–2022	25	55	13	12

**Table 2 cancers-14-04977-t002:** Chemotherapy regimens used in testicular germ cell patients.

Chemotherapy Regimen	Number of Patients (%)
BEP	47 (43)
Carboplatin AUC10	18 (17)
POMB/ACE	11 (10)
Etoposide/Cisplatin (5 days)	11 (10)
Carboplatin AUC7	4 (4)
Escalated Etoposide/Cisplatin (500/60)	9 (8)
Paclitaxel/Etoposide and Paclitaxel/Cisplatin (TE/TP)	4 (4)
Gem/TIP	4 (4)

BEP—Bleomycin/Etoposide/Cisplatin; POMB/ACE—Cisplatin, Vincristine, Methotrexate, Bleomycin/Actinomycin-D, Cyclophosphamide, Etoposide; Gem/TIP—Gemcitabine, Paclitaxel, Ifosfamide, Paclitaxel.

**Table 3 cancers-14-04977-t003:** Complications and adverse events during chemotherapy.

Complication During Chemotherapy	Number of Patients (%)	G1-2	G3 or Above
Neutropenic fever	10 (11)	4	6
Neutropenia	15 (16)	4	11
DVT/PE	9 (10)	0	9
Haemorrhage	1 (1)	1	0
Pulmonary Fibrosis	3 (3)	3	0
Sensory Neuropathy	18 (19)	11	7
Secondary Malignancy	1 (1)	N/A	N/A
Tinnitus	8 (8)	5	3
Heart Failure	1 (1)	0	1
Skin Toxicity	3 (3)	3	0
AKI	3(3)	3	0

AKI—Acute kidney injury; DVT—Deep venous thrombosis; PE—Pulmonary embolism.

## Data Availability

Data is saved onto the hospital electronic notes system.

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
