# Peer review of "A North-West London Experience of the Impact of Treatment Related Toxicity on Clinical Outcomes of Elderly Patients with Germ Cell Tumors"

_cancers, 2022, doi:10.3390/cancers14204977_

Round 1

Reviewer 1 Report

This article aims to understand the clinical outcome after chemotherapy in patients aged over 45 affected by germ cell tumors (GCT) and investigate the possible adverse reactions as a motivation for therapy interruption and therefore a worse clinical outcome. This retrospective observational study analyzes the imaging and the blood samples of 108 patients affected by GCT aged over 45 trying to identify the less disabling but most useful chemotherapy among the different protocols used to treat elderly patients GCTs. The results of the study are evidences that chemotherapy is advantageous in GCS treatment, even in elderly patients, but doses have to be adequated to the age of the patient to avoid adverse effects causing a worse clinical outcome.  

My suggestion:

Title should be changed because in the work there is no mention about COVID 19 impact on chemptherapy so it has to be eliminated

“Clinical outcomes” should be added to keywords given that it’s the aim of the article

The impact of the COVID pandemic should be better investigated because all treatments and follow up have been focibly neglected during the first pandemic wave, so it could have been cause of a worsening of the clinical outcome in some cases. In this regard I suggest you the analysis of this work: https://pubmed.ncbi.nlm.nih.gov/32498612/

There are no exclusion criteria for the patients enrolled. This can represent an important bias fot the study because patients with important comorbilities have a worse compliance for the thrapy and outcome is forcibly negative

The study considerates indiscriminately patients undergone different therapeutic protocols. Data would have been more convincing if the study had been conducted on patients with a standardized and unique therapy

Authors can enlarge the cohort of partients to enroll, giving the work a more relevant importance because the cohort of 108 patients analyzed in the article is too small and not representative for the whole population of over 45 affected by GCTs.

Author Response

Many thanks, I agree with the reviewer to modify the title. We are in the process of doing a Covid 19 paper, and its appropriate that the emphasis on covid 19 should be removed from the paper.

I agree, there will be a bias, as no exlcusion criteria was used. However, we have included all patients >45 who have had chemotherapy at our centre. And this paper is to look at their outcomes, and chemotherapy tolerability. As testicular cancer is found in 15-35, only 10% is in >45. Hence 108 is an adequate number.

Reviewer 2 Report

While the author mentioned in the title that “A north-west London experience of the impact of treatment related to toxicity and COVID 19 on clinical outcomes of elderly patients with germ cell tumors”, I could not find any data related to Covid-19 in this paper?? While some parts of the manuscript are engaging, the quality of the writing is sometimes difficult to follow.

Authors succinctly showed their results mostly in tabular format or in Kaplan Meir curve. It would be really important to show the process of sample selection to final results in flowchart form. For example, starting with year (2003 and 31st March 2022), How many patients authors selected and how they classified them into groups etc., to final outcome in each group

It would be really great (readers/understanding point of view), if authors can show their results in Forest plot/ funnel plot of the relationship between age and treatment regimens etc., Paper looks very sparse just with one Kaplan Meir curve.

Page #5, line #184 insert the reference 

Author Response

I agree with the reviewer, the covid 19 should not be the focus of the paper, this was done by mistake, as the authors have a similar paper in making with a focus on covid 19. We will prepare a flow chart to display the patient selection, as advised by the reviewer.

Reviewer 2

Reviewer 2 comments: While the author mentioned in the title that “A north-west London experience of the impact of treatment related to toxicity and COVID 19 on clinical outcomes of elderly patients with germ cell tumors”, I could not find any data related to Covid-19 in this paper?? While some parts of the manuscript are engaging, the quality of the writing is sometimes difficult to follow

-have deleted the mention to COVID 19 from the paper

Authors succinctly showed their results mostly in tabular format or in Kaplan Meir curve. It would be really important to show the process of sample selection to final results in flowchart form. For example, starting with year (2003 and 31st March 2022), How many patients authors selected and how they classified them into groups etc., to final outcome in each group It would be really great (readers/understanding point of view), if authors can show theirresults in Forest plot/ funnel plot of the relationship between age and treatment regimens etc.,

  • drawn a new Table no 1- to address this issue.
  • Due to the sample size and different cohorts, its not possible to draw a forest plot. I understand the authors suggestion.

Paper looks very sparse just with one Kaplan Meir curve- we have 3 tables and one graph. Germ cell cancer, is a potentially curable malignancy, hence OS/ survival is the most important end point.

Page #5, line #184 insert the reference- done

Reviewer 3 Report

1811898

A north-west London experience of the impact of treatment related toxicity and COVID 19 on clinical outcomes of elderly patients with germ cell tumors

Abstract

Please consider writing testicular germ cell tumour first-time GCT is presented

Keyword : elderly – does this cohort represent an elderly population? (45-70 years)

Introduction

Please consider writing testicular germ cell tumour first-time GCT is presented in the text.

Material

Ethics – does the approval has a number?

It is unclear what kind of data did you collect from electronic records?

Results

needs more structure.

What kind of CT scan performed the images?

 any difference between various types of chemo?

discussion

what kind of limitations has this study?

Age is not a limitation.

The discussion is very limited

Conclusion

Please delete this from the conclusion” Larger scale cohorts may provide further information on the impact of dose adjustments in this patient population”

Author Response

Many thanks, al the suggestions by the reviwer are being incorporated.

The ethics approval was done, with the ethics number- 10215, which is a local audit registration number. CT scan was CT thorax, abdomen and pelvis and this was uniform for all patients. Electronic records- letters, chemotherapy schedule, histopath, tumour marker results.

Limitation sof the study- age, retrospective analysis, and non uniformity of chemotherapy regimes.

Round 2

Reviewer 1 Report

the authors answered all comments and suggestions.

Author Response

Many thanks-

I have inserted a table 1- which has year of diagnosis and demographics based on type of cancer, age distribution and year range.

I cannot insert a forest plot, as the sample size in the patient subgroups is too small to get a good statistical analysis. Reference nymber 22 has been added- PAGE 5- line 184

I have removed the mention of covid 19 in the title.
